# OpenReview forum: "MTRE: Multi-Token Reliability Estimation for Hallucination Detection in VLMs"
_ICLR.cc/2026/Conference — ICLR 2026 Conference Withdrawn Submission_

### Official Review · Reviewer_woL3 · 2025-10-29

**Soundness:** 3
**Presentation:** 3
**Contribution:** 3
**Rating:** 6
**Confidence:** 3

**Summary:**

The paper introduces MTRE, a lightweight white-box method for detecting hallucinations in Vision-Language Models (VLMs). Unlike existing approaches that rely only on the first token's logits, MTRE aggregates information from the first ten tokens using multi-token log-likelihood ratios and self-attention. The method is shown to outperform baseline detectors across multiple benchmarks, including MAD-Bench, MM-SafetyBench, MathVista, and arithmetic tasks.

**Strengths:**

1. This paper proposes a novel hallucination detection criterion, which captures reliability signals through the logits of multiple output tokens.
2. Extensive observational experiments are provided, particularly the analysis and discussion of different types of detection tasks.
3. The proposed method appears to be effective and exhibits promising performance.

**Weaknesses:**

1. As described in the limitations section, the number of open-source VLMs validated by this method is limited. I suggest conducting illusion detection tests on larger-sized VLMs, such as those larger than 32B, to see if the experimental observations also hold true for larger models.
2. Why start with 10 tokens instead of 20? What are the reasons and criteria for this choice?
3. In practice, the proposed method relies to some extent on supervised information. However, in real-world scenarios, data from unseen domains is often more common. Does the proposed method possess any cross-domain capabilities?

**Questions:**

See Weaknesses.

---

> ### Author Response · Authors · 2025-11-29
>
> ## 1. Evaluation on larger VLMs
>
> As described in the limitations section, our original evaluation focused on a subset of open-source VLMs. We appreciate your suggestion to extend the analysis to larger architectures. This is well aligned with our goal of demonstrating that MTRE can adapt to models with different capability levels.
>
> In response, we incorporated experiments on **InternVL3.5-20B** and **LLaVA-NeXT-34B**, together with evaluations on the real-world **MME** [1] and **MMMU** [2] benchmarks. These additions cover both Type-1 and Type-2 response settings. **Section 5.4** of the revised manuscript now includes a motivating explanation reflecting this new evaluation scope.
>
> Across these expanded settings, MTRE continues to yield consistent improvements in hallucination-related failure modes, including those stemming from larger logit spaces and the subtler error patterns exhibited by stronger VLMs.
>
> ## 2. Why start with 10 output tokens? Why not 20?
>
> Due to the largely black-box nature of current VLMs, it is difficult to specify in advance when informative logits first emerge across all tasks and models. We therefore used 10 tokens as an empirically validated starting point.
>
> Although MTRE can operate over longer token sequences, we additionally found that the learned parameter $\tau$ rarely recommends extending the window beyond the first 10 tokens. In response, for the reader's benefit, we additionally add a visualization of the number of output tokens useful for each method **(see Figure 6 in Appendix E.1)**, and often find that $\tau$ is strictly less than or equal to the 10 output tokens used by default in the MTRE procedure.
>
> ## 3. Cross-domain and cross-architecture generalization
>
> Thank you for raising this important point. To evaluate potential generalization beyond the supervised training domain, we conducted cross-architecture transfer experiments: MTRE was trained on the responses of one model and evaluated on another, as well as report on the multi-subject MMMU dataset.
>
> The results, now reported in **Table 17** and **Table 18** in the Appendix, show that MTRE achieves some improved generalization relative to single-token logit-based detectors. Notably, MTRE generalizes particularly well within the MathVista domain, even when trained using responses from a different architecture. Additionally, we note that we select MMMU due to its evaluation on **30 different subjects** to better replicate the expected performance on evaluating unseen subjects.
>
> ### References
>
> [1] Chaoyou Fu, Peixian Chen, Yunhang Shen, Yulei Qin, Mengdan Zhang, Xu Lin, Jinrui Yang, Xiawu Zheng, Ke Li, Xing Sun, Yunsheng Wu, Rongrong Ji, Caifeng Shan, and Ran He. *MME: A comprehensive evaluation benchmark for multimodal large language models*, 2025.
>
> [2] Xiang Yue, Yuansheng Ni, Kai Zhang, Tianyu Zheng, Ruoqi Liu, Ge Zhang, Samuel Stevens, Dongfu Jiang, Weiming Ren, Yuxuan Sun, Cong Wei, Botao Yu, Ruibin Yuan, Renliang Sun, Ming Yin, Boyuan Zheng, Zhenzhu Yang, Yibo Liu, Wenhao Huang, Huan Sun, Yu Su, and Wenhu Chen. *MMMU: A massive multi-discipline multimodal understanding and reasoning benchmark for expert AGI*. In Proceedings of CVPR, 2024.

---

### Official Review · Reviewer_N6aE · 2025-10-29

**Soundness:** 2
**Presentation:** 3
**Contribution:** 2
**Rating:** 4
**Confidence:** 3

**Summary:**

The paper targets VLM hallucination detection and argues that single-token probes (LP, P(True)) miss reliability signals distributed across early token sequences. Empirically, KL divergences between hallucinated vs. truthful continuations grow over several tokens, motivating multi-token analysis. The authors propose MTRE, a lightweight white-box detector that aggregates the first 10 tokens’ logits via multi-token log-likelihood ratios and a small self-attention module; it remains tractable despite large vocabularies.

**Strengths:**

1. The topic is interesting and tries to address an important problem.

2. The paper is well written

**Weaknesses:**

1. The baseline model should add some new models, like LLaVA 1.5, LLaVA NeXT, Qwen 2.5 VL.

2. Figure 1 seems unclear, I recommend the authors add more explanations.

3. For the benchmark discussion, note that several recent studies [1, 2, 3] address both hallucination and maintain performance (even some improvement) on general scenario. I recommend the authors add some benchmarks like OCRBench, MMMU, MME etc.

[1] Mitigating Object Hallucinations via Sentence-Level Early Intervention.

[2] A topic-level self-correctional approach to mitigate hallucinations in mllms.

[3] Rlaif-v: Aligning mllms through open-source ai feedback for super gpt-4v trustworthiness.

**Questions:**

See above.

---

> ### Author Response · Authors · 2025-11-29
> **Adding newer baseline models**
>
> Thank you for your valuable recommendations. We address your comments below.
>
> ## 1. Adding newer baseline models and broader benchmarks
>
> We appreciate your suggestion to include additional recent VLMs such as LLaVA-1.5, LLaVA-NeXT, and Qwen2.5-VL, as well as benchmarks like OCRBench, MMMU, and MME. We agree that broadening the evaluation scope strengthens the empirical validity of MTRE, specifically for more recent models.
>
> In response, we have conducted additional experiments using **InternVL3.5-20B** and **LLaVA-NeXT-34B**, both of which are strong recent models representative of the architectures you recommended. We also incorporated evaluations on **MME** [1] and **MMMU** [2], covering **both Type-1 and Type-2** response settings. We report our new findings in **Section 5.4**. We find that MTRE also provides additional performance improvement over the newer and larger models, showing generalization capabilities across VLM generations.
>
> Furthermore, **we have reviewed and added the referenced recent studies [3, 4, 5] to our related work section** to improve completeness and ensure that our discussion reflects the current literature. We thank you for your suggestion to help bring our work up to current literature.
>
> ## 2. Clarifying Figure 1
>
> Thank you for pointing out that the previous version of Figure 1 lacked clarity. **We have revised the caption to provide a clearer explanation**:
>
> > *"Summary of experiments on MAD-Bench and MM-SafetyBench (2 detection-task types × 4 VLMs × 3 prompts per dataset question). Each cell reports the fraction of experimental settings where the method in the row outperforms the method in the column, measured by Accuracy and AUROC, respectively."*
>
> This updated caption explains the experimental structure and clarifies how to interpret each cell.
>
> ### References
>
> [1] Chaoyou Fu, Peixian Chen, Yunhang Shen, Yulei Qin, Mengdan Zhang, Xu Lin, Jinrui Yang, Xiawu Zheng, Ke Li, Xing Sun, Yunsheng Wu, Rongrong Ji, Caifeng Shan, and Ran He. *MME: A comprehensive evaluation benchmark for multimodal large language models*, 2025.
>
> [2] Xiang Yue, Yuansheng Ni, Kai Zhang, Tianyu Zheng, Ruoqi Liu, Ge Zhang, Samuel Stevens, Dongfu Jiang, Weiming Ren, Yuxuan Sun, Cong Wei, Botao Yu, Ruibin Yuan, Renliang Sun, Ming Yin, Boyuan Zheng, Zhenzhu Yang, Yibo Liu, Wenhao Huang, Huan Sun, Yu Su, and Wenhu Chen. *MMMU: A massive multi-discipline multimodal understanding and reasoning benchmark for expert AGI*. In Proceedings of CVPR, 2024.
>
> [3] Tianyu Yu, Haoye Zhang, Qiming Li, Qixin Xu, Yuan Yao, Da Chen, Xiaoman Lu, Ganqu Cui, Yunkai Dang, Taiwen He, Xiaocheng Feng, Jun Song, Bo Zheng, Zhiyuan Liu, Tat-Seng Chua, and Maosong Sun. *RLAIF-V: Open-source AI feedback leads to super GPT-4V trustworthiness*.
>
> [4] Lehan He, Zeren Chen, Zhelun Shi, Tianyu Yu, Jing Shao, and Lu Sheng. *Systematic reward gap optimization for mitigating VLM hallucinations*, 2025.
>
> [5] Shangpin Peng, Senqiao Yang, Li Jiang, and Zhuotao Tian. *Mitigating object hallucinations via sentence-level early intervention*, 2025.

---

### Official Review · Reviewer_nnVM · 2025-11-01

**Soundness:** 3
**Presentation:** 3
**Contribution:** 3
**Rating:** 4
**Confidence:** 5

**Summary:**

This paper addresses the hallucination detection issue in Vision-Language Models (VLMs) by proposing a novel method called Multi-Token Reliability Estimation (MTRE) and its variant, MTRE-$\tau$ . Departing from prior work of prioritizing the first generated token's logit, MTRE leverages the distributional shift across a sequence of early generated logits to extract richer diagnostic signals. The authors design a reliability classifier, $f_{\theta}$, trained on token-level labels, and aggregate its log-likelihood ratios (LLRs) over a dynamically determined or fixed number of early tokens. Experiments demonstrate MTRE's competitive performance, especially in scenarios where hallucinations emerge late in the sequence (Type II setting).

**Strengths:**

Clear Motivation and Rationale: The fundamental premise—that the full sequence of early logits contains more diagnostic information than the single first token—is clearly articulated and well-supported by preliminary analysis (Section 3).

Sound Empirical Insights: The paper provides several valuable empirical observations (Section 3 and 5) that are useful for the VLM reliability community. For instance, the finding that hallucination divergence may emerge late in the sequence, and consequently, that single-token probing methods (e.g., Zhao et al., 2025) are suboptimal for Type II settings, is a significant takeaway.

Comprehensive Experimental Setup and Clarity: The authors provide a sufficiently detailed description of the experimental setup, particularly in the Appendix regarding model training and hyper-parameter choices. The experimental validation is thorough across various VLM architectures and hallucination settings (Type I and Type II).

Solution to Core Hyper-parameter Identification: The introduction of the MTRE-$\tau$ variant (Section 4.3), which employs cross-fitting for prior estimation (minimizing token-broadcast binary cross-entropy) and dynamic evidence length determination ($\tau_{s_{i}}$), offers a practical approach to tackling the sensitivity of the aggregation step to hyper-parameters.

**Weaknesses:**

Major Concerns

1. Limited Efficacy in Type I Setting: While MTRE significantly outperforms baselines in the Type II setting, its performance edge over the competitive Linear Probing (Lin. Prb.) baseline on Type I tasks (more common ones) is often marginal.

2.  Incremental Gain of MTRE-$\tau$: MTRE-$\tau$ fails to demonstrate a clear and significant performance improvement over MTRE. Given that MTRE-$\tau$ introduces substantial additional complexity (cross-folding, parameter calibration, optimization for $C_{u}$ and $C_{b}$), the marginal utility of this variant is questionable.

3. Extra Computational Cost Analysis: The MTRE method inherently requires training an auxiliary reliability classifier $f_{\theta}$, which increases complexity relative to post-hoc methods. While some performance gain justifies this, the paper lacks a detailed comparative analysis of the computational overhead.  A clearer comparison of the inference time complexity of MTRE against Lin. Prb. needs to be provided.

Minor Concerns

1. The labels OE and OEH in Figure 2 are not immediately clear from the main body text or caption. It would be better to explicitly explain  these abbreviations in the figure caption or the accompanying paragraph.

2. The definition of the $\sigma(\cdot)$ function used in the prior estimation loss (5) should be stated in Section 4.3 (though it is provided in the Appendix)

**Questions:**

Regarding the inference time reported in Table 3 (0.944 ms), please clarify whether this average inference time is measured per single generated token or per complete Q&A query (statement). Even if it is per complete query, the time consumption is highly significant, potentially negating the benefits for real-time deployment or subsequent hallucination mitigation steps (e.g., new inference, re-prompting, or post-hoc corrections).

---

> ### Author Response · Authors · 2025-11-29
>
> Thank you for the insightful feedback. We believe your suggestions have helped strengthen our work. Below, we address each concern and summarize the additions made to the paper.
>
> ## Limited Efficacy in Type I Setting
>
> To clarify, we observe from the token-wise divergence patterns in Figures 2 and 3 that the discriminative signal separating hallucinated from non-hallucinated responses in Type I tasks is already strongly encoded in the model’s initial logits, leaving limited room for MTRE to deliver large absolute improvements. This observation aligns with our hypothesis and with your observation: MTRE’s relative advantages become more pronounced precisely when the discriminative structure is weaker or more diffuse, as is the case in the more challenging Type II tasks, where MTRE consistently yields substantially larger gains.
>
> ## Extra Computational Cost Analysis and Gain of MTRE-$\tau$
> We agree on the suggested additions to improve the transparency of computational cost and marginal utility between MTRE and MTRE-$\tau$. First, we note that the MTRE procedure is classifier-agnostic. When even lower latency is desired, MTRE (LP) employs a lightweight linear probe as the reliability classifier with only 32K parameters **(8.37 MB VRAM, 0.041 ms per detection)** while maintaining strong performance (see Tables 3 and 4). In response, **we add the above raw computational overhead comparison between the two classifiers (Linear Probing and Attention) in Table 5**. Regarding the 0.944 ms inference time reported in Table 3, this value refers to a single inference call of the attention-based classifier. Additionally, we note that MTRE is fully parallelizable across tokens due to the independence of each $z_{s_i, t}$ computation, and users may also substitute a linear classifier when latency is more important. **The caption of Table 5 has been updated to make note of these capabilities**.
>
> We additionally clarify that the primary contribution of MTRE-$\tau$ is not performance alone, but its ability to achieve competitive results with substantially lower hyperparameter search cost, enabled by focusing on fewer tokens through a **shorter evidence length**. In response, **we have added Appendix E.1, Figure 7**, which presents an experiment showing that MTRE-$\tau$ **removes the need for large token-count sweeps** by learning adaptive calibration, thereby reducing search complexity. The results demonstrate that MTRE-$\tau$ dominates the quality–compute Pareto frontier, achieving AUC $\geq$ 0.70 with 1.64× lower per-trial GPU hours (0.035 vs. 0.058) and maintaining this advantage across the full optimization landscape.
>
> ## Ambiguity of OE and OEH Labels
>
> We revised the caption of Figure 2 to explicitly define the **OE** (Open Ended) and **OEH** (Open Ended with Hint) labels to avoid confusion.
>
> ## Definition of $\sigma(\cdot)$
>
> We updated Section 4.3 to reference the binary cross-entropy definition provided in the Appendix, ensuring that the use of $\sigma$ is clear upon first introduction.

---

### Official Review · Reviewer_fd2b · 2025-11-01

**Soundness:** 3
**Presentation:** 3
**Contribution:** 3
**Rating:** 6
**Confidence:** 3

**Summary:**

This paper proposes MTRE (Multi-Token Reliability Estimation), a novel white-box hallucination detection framework for vision-language models (VLMs). Unlike existing single-token probing or P(True) methods that focus on the first generated token, MTRE aggregates information from the logits of multiple early tokens (typically the first 10) to estimate model reliability. By leveraging sequential log-likelihood ratios, attention-based classifiers, and calibration via cross-fitting, MTRE captures reliability dynamics that unfold across token generation. Experimental results on MAD-Bench, MM-SafetyBench, MathVista, and several arithmetic reasoning datasets show consistent gains over strong baselines like linear probing and TokenSAR.

**Strengths:**

1. The paper identifies a key limitation of previous approaches that rely solely on the first-token logit, showing through KL divergence analysis that hallucination-related divergence often arises in later tokens. This motivates the multi-token design in a theoretically grounded way.

2. MTRE introduces a lightweight yet principled multi-token aggregation method, formulated as a calibrated sequential log-likelihood ratio test. The design effectively balances interpretability and computational efficiency.

3. The authors conduct evaluations on four open-source VLMs (LLaVA-v1.5, mPLUG-Owl, LLaMA-Adapter V2, MiniGPT-4) and multiple benchmarks. The results demonstrate robust and consistent improvements, including on challenging self-evaluation (Type 2) settings.

4. MTRE adds negligible overhead (<1% VRAM and inference time) and does not require retraining large models, making it feasible for deployment in real-world systems.

**Weaknesses:**

1. The current comparison is restricted to models such as LLaVA-v1.5 (7B), mPLUG-Owl, LLaMA-Adapter V2, and MiniGPT-4, which may now be considered relatively early-generation VLMs. It remains unclear how MTRE performs on more recent and stronger models (such as LLaVA-Next, InternVL2 or Qwen2.5-VL), which exhibit lower hallucination rates. Incorporating these models would better demonstrate the generality and contributions of MTRE.

2. All experiments use 7B-scale models. It would be informative to analyze whether MTRE scales effectively to larger architectures (e.g., 13B) or smaller lightweight models (3B in scale).

3. The current datasets primarily involve English and relatively simple visual reasoning. Evaluating on multilingual or real-world datasets (e.g., OCR-heavy or video-based tasks) could further validate robustness.

**Questions:**

1. Include Stronger Baselines: Evaluate MTRE on newer VLMs such as LLaVA-Next, Qwen2.5-VL, or InternVL3 to verify whether multi-token reliability estimation remains beneficial when hallucinations are rarer but more subtle.

2. Ablation Across Model Sizes and Modalities: Analyze how performance scales with model size and whether similar gains appear for larger or multilingual models.

---

> ### Author Response · Authors · 2025-11-29
> **Stronger Baselines: Evaluation on newer VLMs across model sizes and modalities**
>
> Thank you for your time in reviewing our work. As you correctly noted, our aim is to introduce a lightweight multi-token aggregation method formulated as a calibrated sequential log-likelihood ratio test for hallucination detection in VLMs. We address your suggestions below, as well as include a revision of our paper (edits marked in red).
>
> ## Stronger Baselines: Evaluation on newer VLMs across model sizes and modalities
>
> We appreciate your recommendation to examine MTRE on more recent and higher-capacity VLMs. This aligns closely with our goal of demonstrating that multi-token reliability estimation remains beneficial across different model capability regimes.
>
> In response, we expanded our evaluation to include **InternVL3.5-20B** and **LLaVA-NeXT-34B**, two widely available and strong open-source VLMs. To further stress-test cross-modal and reasoning robustness, we additionally incorporated the **MME** [1] and **MMMU** [2] benchmarks, covering **both Type-1 and Type-2** response settings.
>
> As reported in **Section 5.4**, MTRE continues to provide consistent improvements on hallucination-related failures across these models and tasks. Notably, MTRE remains effective even when applied to (1) larger logit spaces, (2) higher-performing VLMs, and (3) subtler error patterns that arise at scale. We added a clear motivating explanation to Section 5.4 to reflect this enhancement to the evaluation scope.
>
> Your suggestion to analyze performance scaling across model sizes and modalities is well taken. The newly added experiments across 20B–34B architectures serve as an initial step in this direction, showing that MTRE's gains persist as model capacity increases. Additionally, results on **multilingual and difficult domain-specific benchmarks** such as MME and MMMU provide evidence that MTRE is able to generalize across multiple input domains.
>
> ### References
>
> [1] Chaoyou Fu, Peixian Chen, Yunhang Shen, Yulei Qin, Mengdan Zhang, Xu Lin, Jinrui Yang, Xiawu Zheng, Ke Li, Xing Sun, Yunsheng Wu, Rongrong Ji, Caifeng Shan, and Ran He. *MME: A comprehensive evaluation benchmark for multimodal large language models*, 2025.
>
> [2] Xiang Yue, Yuansheng Ni, Kai Zhang, Tianyu Zheng, Ruoqi Liu, Ge Zhang, Samuel Stevens, Dongfu Jiang, Weiming Ren, Yuxuan Sun, Cong Wei, Botao Yu, Ruibin Yuan, Renliang Sun, Ming Yin, Boyuan Zheng, Zhenzhu Yang, Yibo Liu, Wenhao Huang, Huan Sun, Yu Su, and Wenhu Chen. *MMMU: A massive multi-discipline multimodal understanding and reasoning benchmark for expert AGI*. In Proceedings of CVPR, 2024.

---

### Note · Authors · 2026-01-26

I have read and agree with the venue's withdrawal policy on behalf of myself and my co-authors.

---

### Meta-Review · Area_Chair_VKD3 · 2026-01-05

**Summary:**

The paper proposes Multi-Token Reliability Estimation (MTRE), a white-box method for detecting hallucinations in Vision-Language Models (VLMs). The core premise is that hallucination signals may not be fully captured by the first token alone, but rather emerge over the sequence of early logits. The authors propose training a lightweight classifier (either attention-based or linear) on the logits of the first 10 tokens to distinguish between reliable and hallucinated outputs. While the reviewers appreciated the intuitive motivation and the authors' responsiveness during the rebuttal period, the consensus leans towards rejection. The primary grounds for this decision are the marginal performance gains in standard settings relative to the added complexity. While the authors effectively addressed concerns regarding the age of the base models by incorporating InternVL3.5-20B and LLaVA-NeXT-34B, the method's performance on Type I (direct answering) tasks remains comparable to much simpler baselines like single-token Linear Probing. The reviewers found that the significant gains were largely restricted to Type II (self-evaluation) tasks. Furthermore, the proposed variant, MTRE-$\tau$, introduces additional algorithmic complexity (cross-fitting and calibration) without yielding a decisive performance advantage over the base MTRE method, raising questions about its practical utility. Given the high bar for ICLR, the trade-off between the method's white-box requirements/training overhead and the incremental gains in general scenarios is not sufficiently favorable at this time.

**Reviewer Concerns:**

Addressed Concerns:
* Outdated Base Models: Reviewers fd2b, N6aE, and woL3 all expressed concern that the original submission relied on older 7B models (LLaVA-v1.5, MiniGPT-4). The authors successfully addressed this by conducting new experiments on InternVL3.5-20B and LLaVA-NeXT-34B, demonstrating that the method functions on larger, modern architectures.
* Benchmark Coverage: Reviewer N6aE requested broader benchmarking. The authors addressed this by adding results on MME and MMMU.
* Computational Overhead: Reviewer nnVM requested a clearer analysis of inference costs. The authors provided a breakdown showing the overhead is minimal relative to VLM decoding, though the necessity of the auxiliary training stage remains a structural cost.

Outstanding Concerns:
* Marginal Gains on Type I Tasks: Reviewer nnVM highlighted that for direct answering tasks (Type I), the method performs similarly to single-token linear probing. The authors acknowledged this, stating that discriminative signals are strong in the first token for these tasks. However, this limitation significantly narrows the scope of the paper's impact, as Type I is the primary deployment mode for VLMs.
* Utility of MTRE-$\tau$: Reviewer nnVM questioned the marginal utility of the complex MTRE-$\tau$ variant. While the authors argued it aids hyperparameter search efficiency, the reviewers were not convinced that this justifies the added complexity of cross-fitting and calibration compared to the simpler MTRE or baseline methods.
* White-Box Dependency: While inherent to the method, the requirement for full logit access to the first 10 tokens limits applicability compared to black-box or single-token methods, particularly when the performance gap is narrow.

**Reviewer Scores:**

* Reviewer fd2b (Score: 6): This reviewer leaned positive and their main concern regarding model scale was addressed. However, they noted the paper was "marginally" above the threshold. Had they participated fully in the discussion regarding the marginal Type I gains, they likely would have maintained their score or dropped slightly to a 5, acknowledging the improvements but recognizing the limitations in broad applicability.
* Reviewer nnVM (Score: 4): This reviewer would likely maintain their score. Their primary concern regarding the limited efficacy in Type I settings and the questionable value-add of the MTRE-$\tau$ variant was confirmed by the authors' response rather than refuted. The acknowledgment that first-token signals are sufficient for Type I tasks reinforces the reviewer's skepticism about the necessity of the proposed multi-token complexity for general use cases.
* Reviewer N6aE (Score: 4): This reviewer would likely have maintained a 4. While the authors added the requested benchmarks (MME, MMMU), the reviewer rated the contribution as "fair" and soundness as "fair." The rebuttal improved the evaluation scope but did not fundamentally alter the mechanics or the limited impact of the method on general hallucination scenarios compared to simpler baselines.
* Reviewer woL3 (Score: 6): This reviewer was generally positive about the novelty. The clarification regarding the choice of 10 tokens and cross-domain capabilities likely solidified their score. However, they largely focused on observational experiments rather than the comparative marginal utility that concerned Reviewer nnVM.

---

### Decision · Program_Chairs · 2026-01-26

Reject